# Extending the dynamic range of biomarker quantification through molecular equalization

Sharon S. Newman [1,2,7], Brandon D. Wilson [3,4,7], Daniel Mamerow[2,4,7], Benjamin C. Wollant[2], Hnin Nyein[4], Yael Rosenberg-Hasson[5], Holden T. Maecker [5], Michael Eisenstein[2,4] & H. Tom Soh [2,4,6] ✉

Precision medicine requires highly scalable methods of multiplexed biomarker quantification that can accurately describe patient physiology. Unfortunately, contemporary molecular detection methods are generally limited to a dynamic range of sensitivity spanning just 3–4 orders of magnitude, whereas the actual physiological dynamic range of the human plasma proteome spans more than 10 orders of magnitude. Current methods rely on sample splitting and differential dilution to compensate for this mismatch, but such measures greatly limit the reproducibility and scalability that can be achieved—in particular, the effects of non-linear dilution can greatly confound the analysis of multiplexed assays. We describe here a two-pronged strategy for equalizing the signal generated by each analyte in a multiplexed panel, thereby enabling simultaneous quantification of targets spanning a wide range of concentrations. We apply our 'EVROS' strategy to a proximity ligation assay and demonstrate simultaneous quantification of four analytes present at concentrations spanning from low femtomolar to mid-nanomolar levels. In this initial demonstration, we achieve a dynamic range spanning seven orders of magnitude in a single 5 μl sample of undiluted human serum, highlighting the opportunity to achieve sensitive, accurate detection of diverse analytes in a highly multiplexed fashion.

To fully realize the vision of precision health, there is a pressing need for highly multiplexed biomarker quantification methods that can accurately describe patient physiology. One of the fundamental challenges of contemporary biomarker quantification technologies is the relatively limited quantifiable range inherent to most signal detection modalities, which typically spans 3–4 orders of magnitude[1]. This poses a difficult challenge when measuring multiple protein biomarkers in blood, given that the dynamic range of concentrations in the plasma proteome spans 10+ orders of magnitude[1]. To achieve quantitation across this full range of concentrations, most current methods separate the sample into different panels for the measurement of sets of proteins that fall within roughly the same concentration range. Each panel is processed to specifically tune the signal output from a given set of analytes to the quantifiable regime of the detector. For panels of high-abundance analytes, samples are typically diluted until the signals generated are below the upper limit of quantitation (ULOQ). For low-

[1]Department of Bioengineering, Stanford University, Stanford, CA 94305, USA. [2]Department of Electrical Engineering, Stanford University, Stanford, CA 94305, USA. [3]Department of Chemical Engineering, Stanford University, Stanford, CA 94305, USA. [4]Department of Radiology, School of Medicine, Stanford University, Stanford, CA 94305, USA. [5]Institute for Immunity, Transplantation, and Infection, School of Medicine, Stanford University, Stanford, CA 94305, USA. [6]Chan Zuckerberg Biohub, San Francisco, CA 94158, USA. [7]These authors contributed equally: Sharon S. Newman, Brandon D. Wilson, Daniel Mamerow. ✉e-mail: tsoh@stanford.edu

abundance analytes, the output signal is amplified to raise the final signal above the lower limit of quantitation (LLOQ). Unfortunately, the need for multiple assay panels also means that larger sample volumes are required, which poses challenges for precious clinical specimens such as bio-banked samples[2].

Critically, the dilution process introduces the notoriously difficult problem of non-linear dilution. This describes the phenomenon wherein measured concentrations of a given analyte deviate greatly from their expected values when measured at different dilutions, thereby undermining meaningful comparisons of measurement results from multiple panels[3,4]. The effects of non-linear dilution can be dramatic—for example, upon comparing undiluted patient serum samples to those that were diluted 3-fold, Rosenberg-Hasson et al. observed that only 6% of the biomarkers exhibited a proportional change in signal upon dilution (Supplementary Fig. 1). Indeed, they observed changes in signal ranging from 0.61- to 5.45-fold, with the signal from some proteins even increasing upon dilution[3]. Most troublingly, the observed effects of non-linear dilution varied not only from analyte to analyte but also from sample to sample, suggesting that the optimal dilution for each target could vary across patients. Although there are methods for assessing the magnitude of a non-linear dilution effect, such as spike-and-recovery assays[5], there is currently no general solution for circumventing the non-linear dilution problem.

In this work, we introduce EVROS (after the Greek word εύρος meaning "range") – an equalization methodology that enables multiplexed quantification of protein biomarkers over a widely-divergent concentration range from a single microliter-scale sample, without differential dilution and amplification. EVROS employs a pair of tuning mechanisms—probe loading and epitope depletion—to individually tune the binding signal generated by each analyte, thereby equalizing their signal output into the same quantifiable dynamic range. As an exemplar, we simultaneously quantify a panel of four proteins in a single 5-μl sample of undiluted human serum for which the physiological concentrations can range from <20 fM (interleukin-6; IL-6) to >200 nM (C-reactive protein; CRP), spanning seven orders of magnitude. In this work, we demonstrate a version of EVROS based on the standard solid-phase proximity ligation assay (spPLA)[6,7], but this method could readily be implemented with a variety of other proximity-based methods (PEA, cPLA, 3PLA, etc)[8–12], and we believe this approach should offer a broadly practical strategy for the design

of sensitive, reproducible, and highly-multiplexed molecular detection assays.

## Results

### Overcoming the dynamic range limits of multiplexed quantitation

Most affinity reagents have similar binding curves that can be modeled by the Langmuir isotherm (Fig. 1a, black dotted line)[13,14]. The quantitative signal resolution of typical detection modalities is limited to 3–4 orders of magnitude (gray bar), which in turn constrains the output signal curve that can be achieved (black solid line). In the hypothetical example shown in Fig. 1a, sufficient resolution would only be achieved for analytes present at intermediate concentrations (red). The signal from lower-abundance analytes (yellow) is typically masked by the background and noise floor of the detection modality, producing little to no detectable signal. In this scenario, the signal output is usually brought up into the quantifiable range by signal amplification strategies[15,16]. On the other hand, the signal from high-abundance analytes (blue) tends to reach the saturation limit of the detection modality, such that changes in concentration do not produce meaningful changes in the signal. Dilution can shift the measured concentration range to produce quantifiable signals that fall within the linear range of the binding curve (red), but this also introduces the unpredictable effects of non-linear dilution, which can lead to incorrect measurements and false conclusions. Additionally, amplification and dilution are challenging to implement simultaneously in a multiplexed assay; dilution further decreases the already-low signal from scarce analytes, whereas signal amplification could push the signal from medium to high-abundance analytes beyond the ULOQ. Thus, the crux of multiplexed analyte measurements of samples across large concentration ranges is to decouple the modulation of response curves for each analyte.

Our EVROS strategy employs two tuning principles that make it possible to modulate the signal output curve of each analyte individually. This makes it possible to bring the output signals from multiple analytes that are present at highly divergent concentrations into the same quantitative regime with good resolution (Fig. 1b). The first tuning mechanism, 'probe loading', effectively shifts the binding curve of our detection reagents to achieve good resolution at the physiological concentration range of the analyte[14]. The second strategy, 'epitope depletion', attenuates signals from higher-abundance analytes to

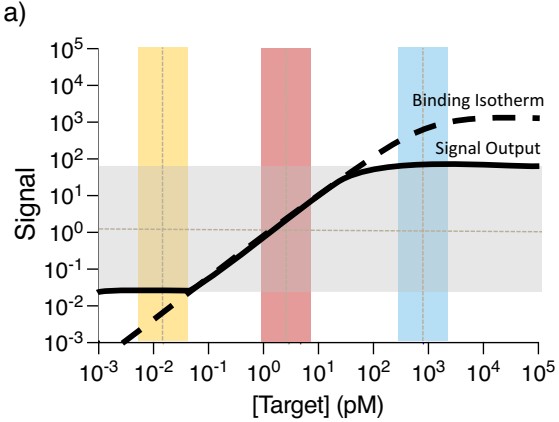

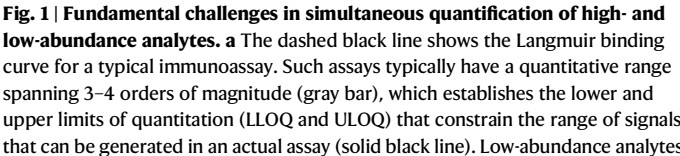

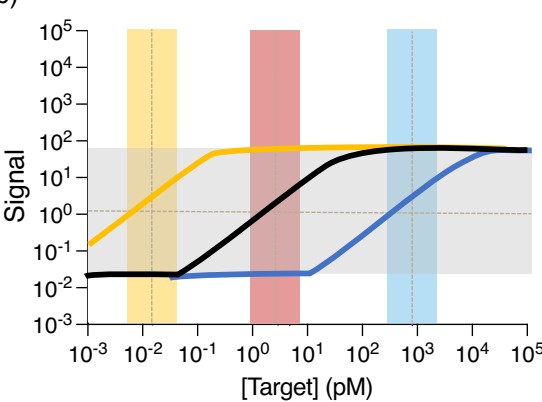

**Fig. 1 | Fundamental challenges in simultaneous quantification of high- and low-abundance analytes. a** The dashed black line shows the Langmuir binding curve for a typical immunoassay. Such assays typically have a quantitative range spanning 3–4 orders of magnitude (gray bar), which establishes the lower and upper limits of quantitation (LLOQ and ULOQ) that constrain the range of signals that can be generated in an actual assay (solid black line). Low-abundance analytes

(yellow bar) typically have low quantitative resolution, generating a signal that falls below the LLOQ, whereas high-abundance analytes (blue bar) produce saturating signals that reach the ULOQ. Only intermediate concentrations (red bar) can be resolved satisfactorily in this scenario. **b** The tuning mechanisms employed in EVROS make it possible to shift the signal response curve to achieve good quantitative resolution at any concentration range.

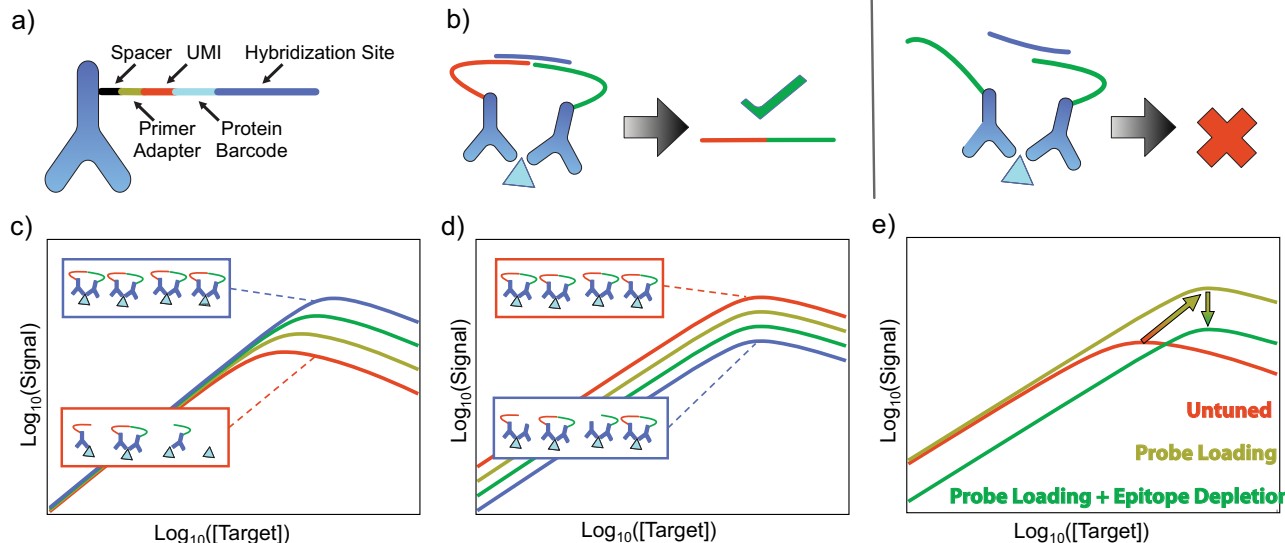

**Fig. 2 | Tuning spPLA-based quantification of low- and high-abundance targets.**
**a** For spPLA, the dAbs are coupled to DNA strands which encode barcodes that are unique for each molecule and antibody pool, as well as a hybridization site.
**b** Signals are produced when two correctly paired dAbs bind to the same molecule (left), such that a splint sequence (purple) can bind to each dAb's hybridization site, producing a DNA reporter. For all other binding events, no signal is generated (right). **c** Our first tuning mechanism, probe loading, entails increasing the dAb concentration to shift the equilibrium and increase the fraction of target molecules that are bound to multiple dAbs. This increases the output for all target concentrations (see probe-loaded blue curve versus untuned red curve). The increase

is more pronounced at higher target concentrations, effectively shifting the response curve up and to the right. **d** The second mechanism, epitope depletion, tunes the reporter output for a given target. One can shift the response curve downward by adding a defined concentration of a depletant antibody, which lacks a DNA tag, thus reducing the likelihood of producing a readout for a given dAb binding event (see depleted blue curve versus untuned red curve). **e** Since epitope depletion on its own produces a weaker signal, we combine this tuning strategy with probe loading, which has the net effect of shifting the binding curve to the right.

ensure that similar outputs are produced for all targets regardless of their abundance.

In principle, these two tuning mechanisms can be applied to many types of proximity-based immunoassays[8–12] based on detection antibodies (dAb) coupled to signaling moieties such as a DNA strand or fluorophore. For the present work, we have implemented EVROS in the context of the solid-phase proximity ligation assay (spPLA)[7,17] due to its low volume requirements, high specificity, and high throughput capabilities. This assay format employs polyclonal antibodies that are divided into three pools for each target—one of which is coupled to magnetic beads and acts as a capture antibody (cAb), and two pools of DNA-labeled dAbs (Fig. 2a, Supplementary Fig. 2). This approach eliminates the need to screen multiple sets of monoclonal antibodies, because polyclonal antibody pools are very likely to simultaneously bind multiple distinct epitopes on a target. When the two dAbs bind to the same captured target molecule, their associated DNA strands undergo a ligation reaction in the presence of a complementary 'hybridization splint' DNA strand and ligase enzyme (Supplementary Fig. 2e), generating a DNA reporter sequence that is subsequently amplified and analyzed via high-throughput sequencing (HTS). These reporters contain protein-specific barcodes and unique molecular identifiers (UMIs)[18] that enable us to assign each target an individual read count based on the sequencing data. One set of dAbs for each target is functionalized with a DNA strand attached by its 5′ end, while the other dAb is coupled to its DNA strand by the 3′ end, such that only target molecules bound by correctly paired 5′- and 3′-dAbs will generate a DNA reporter (Fig. 2b).

Our probe loading tuning mechanism entails changing the concentrations of dAbs for a given analyte such that the signal response is appropriately calibrated to that analyte's physiological abundance. Probe loading shifts the signal response curve up and slightly to the right because of the change in equilibrium based on Le Chatelier's Principle: as the total concentration of dAbs increases, the equilibrium

shifts towards analyte molecules that are bound to a greater number of dAbs (Fig. 2c). This increases output signal asymmetrically with increased analyte concentration, where higher-concentration analytes are more strongly affected by the increased dAb concentration than lower-concentration analytes. The equilibrium model of dAb-target binding is described in detail in Supplementary Note 1 and forms the conceptual framework through which we analyze the two tuning mechanisms. Probe loading is primarily useful in that it increases the signal output for low-abundance analytes, but the subtle rightward shift in the binding curve that it produces can also enable better quantitative resolution for high-abundance analytes.

Our second tuning mechanism, epitope depletion, is used to adjust the signal output produced by high-abundance analytes. Here, we modulate the reporter output for an analyte by controlling the fraction of dAb pairs that result in a signal (Fig. 2d) via the addition of unlabeled 'depletant' antibodies. These are derived from the same pool of antibodies as the dAbs, but do not have signaling moieties. The addition of depletant decreases the probability that an analyte bound to multiple antibodies will produce a signal according to the binomial distribution (Supplementary Note 1). We can modulate the efficiency with which signals are generated by tuning the fraction of depletant antibody, which results in a downward shift of the response curve. Epitope depletion by itself results in a much lower output signal, and so we typically combine this measure with increased probe loading to obtain a readout that falls within the assay's quantitative range.

Probe loading and epitope depletion can be independently applied to individual analytes in the context of a multiplexed assay. This means that in a single sample, we can apply probe loading to shift the response curve for low-abundance analytes upwards, while also using probe loading to shift the binding curve for high-abundance analytes to the right to increase quantitative resolution and implementing epitope depletion to prevent the signal output from reaching saturation (Fig. 2e). This enables high-dynamic-range, multiplexed,

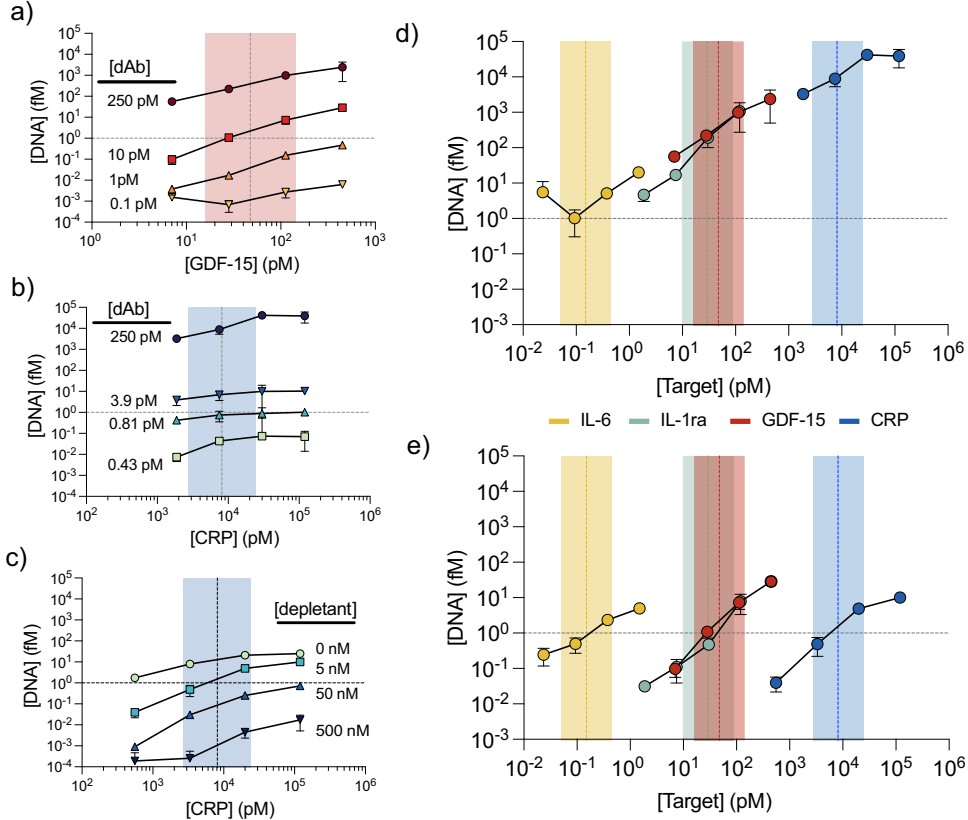

**Fig. 3 | Tuning the detection range for analytes of varying concentration. a** Tuning spPLA output based on quantitative PCR (qPCR) for a low/medium-abundance protein, GDF-15. Decreasing the dAb concentration from a starting point of 250 pM shifts the response curve downwards. We reached our target output of ~1 fM reporter DNA (gray horizontal dotted line) at ~50 pM GDF-15—the log-middle of our target concentration range—with 10 pM dAb. **b** Our two tuning mechanisms can be used in tandem to obtain a sensitive signal response for high-abundance targets like C-reactive protein (CRP). Decreasing probe loading from 250 pM decreases the reporter DNA output at each CRP concentration but does not improve the assay's quantitative sensitivity over the physiological range. **c** By also applying epitope depletion with unlabeled, competing antibodies, we decreased the DNA output to the desired range and shifted the overall response curve to the

right, so that linear detection occurred in the physiologically relevant range. To compensate for the reduced output caused by epitope depletion, we used 3.93 pM dAb, a concentration selected to generate a pre-depletion output of 10 fM reporter. **d, e** Single-plex qPCR readout for four protein analytes before (**d**) and after (**e**) tuning for each individual analyte. It should be noted that each analyte was measured separately in qPCR to have signal resolution for each analyte. For all panels, error bars represent the standard deviation of three qPCR replicates centered around the mean value. For the two lowest concentrations of CRP at 500 nM depletant in panel **c**, the lower error bar value is negative and therefore cannot be plotted on the log-scale y-axis. Colored vertical bars represent the target concentration range. Source data are provided as a Source Data file.

protein quantitation without the need for differential sample processing.

**Experimental demonstration of the two tuning mechanisms**
We first demonstrated our ability to predictably shift analyte response curves with the growth/differentiation factor-15 (GDF-15) protein, which is a biomarker for inflammation, myocardial ischemia, and cancer[19]. Since physiological concentrations can vary as a function of disease state, we designed our testing range (5–500 pM) to be broader than the nominal GDF-15 concentration range of ~5–120 pM (~0.2–5.0 ng/mL) in serum[6,20]. Our goal was to shift the response curve until the DNA reporter output concentration was ~1 fM at the log-middle of the testing concentration range (~50 pM). This 1 fM target was established empirically as the concentration of DNA reporter that maximizes the quantitative precision conferred by the UMIs (Supplementary Fig. 3). To expedite the tuning process, we initially measured the DNA reporter output with quantitative PCR (qPCR) rather than the full sequencing workflow. Starting from standard spPLA conditions of 250 pM of each dAb, we tuned the assay in the buffer by incrementally decreasing the probe concentration. As expected, decreasing dAb reduced the reporter output of the system (Fig. 3a), shifting the response curve downwards. We identified 10 pM as the optimal

concentration for each dAb, yielding an output of ~1 fM DNA at the center of the tested concentration range of GDF-15. This trial-and-error tuning process is relatively straightforward, but would be tedious and time-consuming to perform at scale for larger numbers of targets. We therefore developed heuristics based on empirical trends from past tuning data to rapidly optimize probe concentrations given a desired concentration range and signal output (Supplementary Note 2).

For high-abundance proteins, it is difficult to accurately quantify changes in concentration that occur beyond the ULOQ of the signal response curve (Fig. 1a). The goal here is therefore to shift the curve to the right, to generate a more sensitive signal response over the desired concentration range within the steep-sloped portion of the curve. Incorporating epitope depletion makes this possible—but since epitope depletion by itself would result in a very low output signal, we must combine this measure with increased probe loading to obtain a readout that is of the same order of magnitude as the other targets. We demonstrated this with CRP, a high-abundance target with a basal concentration range of 1–36 nM. At the initial dAb concentration of 250 pM, the reporter DNA output concentration far exceeded our target of 1 fM at the log-middle of the CRP testing range (1–70 nM). Our tuning heuristic correctly predicted that decreasing the dAb concentration to 0.81 pM would commensurately decrease reporter DNA

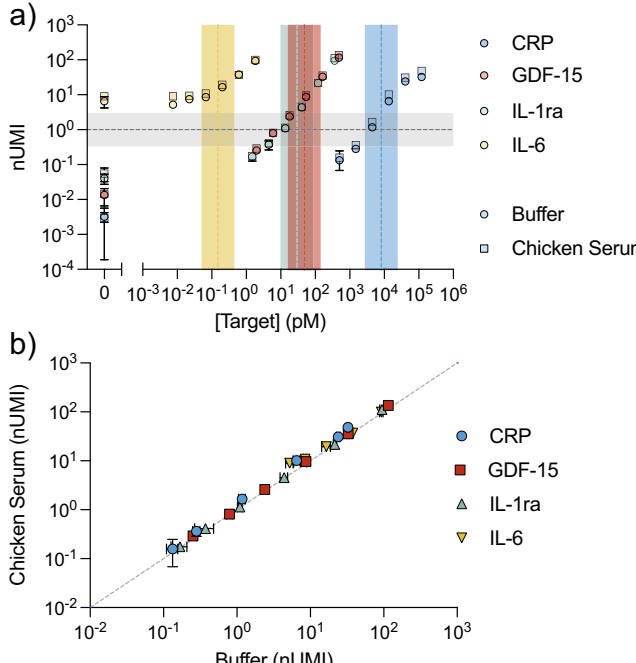

**Fig. 4 | Simultaneous quantification of four protein analytes at concentrations spanning seven orders of magnitude. a** nUMI counts as a function of analyte concentration from a tuned four-plex assay with six standards (Supplementary Table 1) containing different concentrations of our four analytes spiked into buffer and chicken serum. The dotted vertical lines represent the approximate log-middle of the defined testing range for each target. The dotted horizontal line is our target output of ~1 fM reporter DNA. **b** We observed close correlation between the measurements obtained in chicken serum and buffer (linear fit: $m = 1.004 \pm 0.011$, $b = 0.070 \pm 0.011$, $R^2 = 0.992$). For both a and b, all samples were run in triplicate, the plotted data were derived from a single demultiplexed sequencing run, and no background subtraction was applied. Error bars represent the standard deviation of three replicates centered around the mean value. Source data are provided as a Source Data file.

output to 1 fM (Fig. 3b). However, this probe tuning step alone was insufficient to achieve quantitative resolution of CRP over the desired concentration range and produced a very flat response curve. We therefore applied epitope depletion and probe loading together to steepen the signal-response curve while also shifting it to the right. We determined that adding 50 nM of depletant would result in a roughly 10–15-fold drop in DNA output signal (Supplementary Fig. 4), and therefore used our heuristic to identify an initial probe loading concentration at which the predicted reporter output would be 10-fold higher: 10 fM rather than 1 fM. We determined that a 3.93 pM concentration of each dAb would appropriately compensate for the reduced reporter output due to epitope depletion (Fig. 3b). After testing a range of depletant antibody concentrations (Fig. 3c), we determined that 5 nM depletant produced the optimal signal response, demonstrating that the use of both tuning mechanisms in tandem makes it possible to generate a highly sensitive response to high-abundance analytes in buffer. It should be noted that the quantification of high-abundance analytes in proximity-based assays is often plagued by the "hook effect", in which high target concentrations lead to decreased signal;[21] we discuss this phenomenon and how we evaded it in Supplementary Note 3.

### Four-plex quantification of analytes at highly divergent concentrations in undiluted chicken serum

The tuning mechanisms described above make it possible for us to achieve the quantification of multiple targets with concentrations

spanning a vast dynamic range in a single measurement. As a demonstration, we performed a four-plex EVROS strategy to spPLA to simultaneously measure CRP, GDF-15, interleukin-1 receptor agonist (IL-1ra), and IL-6 spiked into buffer and serum at physiologically relevant concentrations. The basal physiological ranges of these proteins collectively span seven orders of magnitude (Supplementary Table 1). In an untuned single-plex sp-PLA assay with a qPCR readout, we found that the DNA output concentrations from these four targets spanned more than five orders of magnitude across the tested target concentrations, with especially poor resolution for high-abundance analytes (Fig. 3d). Consequently, if all analytes were read simultaneously, the HTS signal from CRP would drown out that from IL-6 in an untuned assay, with CRP accounting for greater than 99.99% of sequencing reads.

Using our tuning heuristics, we were able to quickly determine the optimal probe and depletant concentrations that would produce reporter DNA output for all targets within a range spanning three orders of magnitude (Fig. 3e, Supplementary Table 2). As expected, the background signal increases with probe loading, but the signals produced by the various analytes at their physiological concentrations are still higher than the respective background signal (Supplementary Fig. 5, Fig. 4a). This tuning process conferred the capability to measure all targets in the desired concentration range simultaneously using a sequencing-based readout. To demonstrate this, we prepared six standards spanning the full testing range of concentrations for the four analytes (Supplementary Table 3). Initial experiments were conducted in 50 μL reactions: 5 μL of buffer plus 45 μL of assay reagents (standard containing spiked targets, dAbs, hybridization splint, etc.).

EVROS-tuned spPLA consistently produced quantitative binding curves for all four analytes in assays performed with all six standards in buffer (Fig. 4a, circles). In this multiplexed assay, we employed an HTS readout rather than qPCR. Instead of detecting DNA reporter concentration, we report normalized UMI (nUMI) counts—the number of UMIs associated with an analyte normalized to the number of UMIs for a control oligo. This normalization accounts for variabilities, such as differences in library pooling. As expected, the signal outputs for all analytes were localized within a range spanning three orders of magnitude, such that none of the analytes' signals drowned out those from the other analytes. As such, we were able to measure low femtomolar concentrations of IL-6 and high nanomolar concentrations of CRP in a single sample. Importantly, these results exhibited excellent target specificity, as we would predict for an spPLA-based assay: when we spiked individual analytes into reactions containing detection reagents for all four analytes, the assay consistently generated an appropriate analyte-specific readout with no measured off-target signal (Supplementary Fig. 6). With the current targets, we demonstrated the quantification of protein concentrations spanning more than seven orders of magnitude, from 8 fM IL-6 to 122 nM CRP (Supplementary Table 1).

Complex biological matrices such as serum contain abundant interferents that can greatly impair the performance of immunoassays relative to results obtained in buffer. We therefore tested whether spPLA tuned with EVROS can obtain high-resolution measurements in undiluted chicken serum. We chose this medium because antibodies against human proteins typically do not cross-react with their chicken homologues, thereby minimizing the confounding effects of endogenous proteins in the serum sample[6]. Using the same optimized reaction conditions, we again measured the same six standards described above—with 5 μL undiluted chicken serum instead of buffer—to produce binding curves for all four analytes (Fig. 4a, squares). We observed highly consistent nUMI values for each analyte in both buffer and chicken serum (linear fit, $R^2 = 0.992$), indicating the assay's robustness in complex sample matrices (Fig. 4b). The slope of $1.004 \pm 0.011$ and y-intercept of $0.070 \pm 0.011$ further highlight the strong concordance between assay measurements collected in buffer and serum.

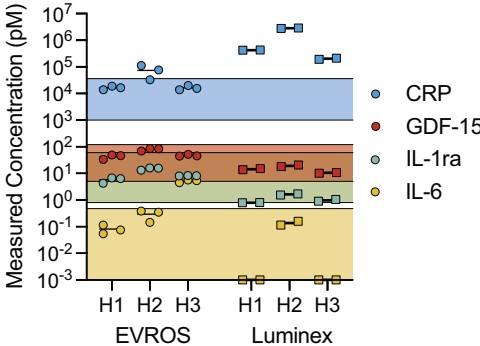

**Fig. 5 | Quantification of endogenous concentrations of four biomarkers in human serum samples.** Three individual serum samples from healthy male donors were measured for IL-6, IL-1ra, GDF-15, and CRP using our EVROS tuning strategy on a four-plex spPLA assay (left) and three separate Luminex assays (right). The shaded bars represent the approximate physiological ranges expected in healthy donors. The EVROS-tuned assay was performed on 5 μL of undiluted serum, whereas the Luminex assays were performed on 25 μL of serum diluted according to the manufacturer's recommendations: 1,000-, 100-, and 3-fold for CRP, GDF-15, and IL-6/IL-1ra, respectively. Bars represent means based on separate measurements of each sample–triplicate for EVROS and duplicate for Luminex. Source data are provided as a Source Data file.

## Four-plex quantification of analytes in undiluted human serum

To demonstrate the utility of our tuning method for quantification in human samples, we measured endogenous concentrations of IL-6, IL-1ra, GDF-15, and CRP in undiluted human serum samples. We replicated the tuned reaction conditions above, using 5 μl of serum samples from three anonymous donors (see Methods) combined with 45 μl of spPLA assay reagents. After sequencing and obtaining nUMI counts for each target in the sample, we calculated their respective concentrations using binding curves developed with the six standards in buffer (see Methods, Supplementary Fig. 7). We were able to directly detect and quantify IL-6, IL-1ra, GDF-15, and CRP simultaneously in all three samples of undiluted human serum (Fig. 5, left). In parallel, we performed a separate experiment to confirm that measurements with these same samples would have been heavily confounded by nonlinearity if diluted for spPLA (Supplementary Fig. 8). These results exhibited considerable deviation both across dilutions as well as across samples, highlighting the clear advantage of employing a non-dilution-based strategy such as EVROS to equalize analyte readouts. The measured endogenous target concentrations for all four analytes were consistent with previously reported physiological values in all three samples. To the best of our knowledge, this represents the first time such a broad range of endogenous analyte concentrations–ranging from low femtomolar IL-6 levels to nanomolar concentrations of CRP–has been detected in a single immuno-assay in human serum without differential dilution.

To compare our measurements to an existing gold-standard method for biomarker quantification, we also measured the four analytes in each serum sample using Luminex panels (Fig. 5, right; Supplementary Table 4). Since Luminex does not offer or recommend a single panel for measuring all four analytes simultaneously, we measured each sample with three different Luminex panels: one for CRP, one for GDF-15, and a 2-plex panel for IL-1ra and IL-6. Notably, Luminex recommends different dilutions for each panel: 1,000-, 100-, and 3-fold for the CRP, GDF-15, IL-1ra/IL-6 panels, respectively. Although measurements for GDF-15 and IL-1ra were consistent between EVROS-tuned-spPLA and Luminex, the Luminex panels encountered problems in measuring the other analytes. For instance, at the recommended dilution for IL-6, two of the three human serum samples were below the limit of detection with Luminex. In contrast,

the EVROS-tuned assay yielded clear measurements of IL-6 for all three serum samples, with two falling within the expected physiological concentration range and the third reading above the expected range. Furthermore, the concentrations of CRP reported by the Luminex assay were unrealistically high; to the best of our knowledge, the measured range of ~200 nM-2.8 μM is near or above the highest reported clinical concentration of CRP observed in end-stage renal failure patients (433 nM)[22,23]. Given that our samples were collected from healthy donors, this is likely due to inaccurate measurements by the Luminex test, and the results from the EVROS-based assay showed that CRP levels were marginally elevated in just one of the three samples. Overall, our results demonstrate that EVROS-based assay tuning can produce robust, multiplexed measurements of analytes in undiluted biological specimens across a remarkably broad concentration range, with performance matching or exceeding that of existing gold-standard approaches.

## Discussion

In this work, we describe EVROS, an equalization methodology that enables multiplexed quantification of proteins with vastly different physiological concentrations in a single, microliter-scale sample without the need for differential dilution. We achieve this by employing a pair of tuning mechanisms–probe loading and epitope depletion–to individually modulate the signal generated by each analyte, allowing equalization of signal output and thus simultaneous quantification of multiple targets across a very large dynamic range. Although these approaches have been employed to some extent in industry settings, to our knowledge, this work represents the first published demonstration of integrating these strategies to tune the dynamic range of a multiplexed immunoassay and overcome the confounding effects of nonlinear dilution. We demonstrate this process for four targets with physiological concentrations spanning more than seven orders of magnitude, from low-femtomolar IL-6 to mid-nanomolar CRP. While we illustrate these tuning principles via spPLA with an HTS readout, the tuning mechanisms we have described are theoretically extensible to any other proximity-based assay[8,11,12,24]. Furthermore, we have not yet encountered any fundamental limitations to the dynamic range that can potentially be achieved with this tuning procedure. Critically, EVROS maintained robust performance in undiluted chicken and human serum, achieving results that are comparable to those obtained in buffer, with performance that was comparable (or even superior, for some analytes) to that of the commercially available Luminex assay.

The equalization methodology of EVROS overcomes several critical technical and scaling problems found in conventional assays. First, by obviating the need for sample dilution, we eliminate the unpredictable but consistently detrimental effects of non-linear dilution that have been observed with other assay formats. This leads to not only potentially more reliable measurements but also eliminates the need for time-consuming processes such as linearity-in-dilution protocols. Instead, for each new analyte added to tune with EVROS, our heuristic can be used to rapidly determine the optimum dAb concentration for that target. Secondly, EVROS provides fine-tuned control over signal production and resolution, which enables efficient use of the quantitative range of assays. For example, although HTS has remarkably high sensitivity and resolution, it is limited by the number of reads available on a flow cell. With EVROS, we can maximize the use of each read in a single run to balance quantitative resolution with the number of analytes and samples desired, minimizing both cost and time in a single run through the assay.

The implementation of EVROS demonstrated here with spPLA and HTS requires only 5 μl of the sample, regardless of the number and abundance of analytes, bringing high-dynamic-range quantitative multiplexing to small volume measurements in a scalable fashion. This low volume requirement unlocks critical applications in the context of scarce and precious clinical samples, such as neonatal care or bio-

banked samples. In these and other contexts, EVROS has the potential to enable more accurate monitoring of multiple targets spanning large concentration ranges in a small sample volume. More frequent collection of tiny blood samples should facilitate biomarker discovery and real-time longitudinal monitoring and characterization of complex disorders and physiological processes, thereby accelerating progress in the realm of precision medicine.

## Methods

### Reagents

All polyclonal antibodies were purchased affinity-purified from R&D Systems. Biotinylated and unbiotinylated antibodies were purchased for CRP (BAF1707 & AF1707), IL-6 (BAF206 & AF206), and IL-1RA (BAF280 & AF280). Unbiotinylated antibodies were purchased for GDF-15 (AF957) and GFP (AF4240). The GDF-15 and GFP antibodies were biotinylated using an EZ-Link Micro NHS-PEG4-Biotinylation Kit from Thermo Fisher Scientific (21955) according to the manufacturer's instructions. The following recombinant target proteins were purchased from R&D Systems: CRP (1707-CR), IL-6 (206-IL), IL-1RA (280-RA), and GDF-15 (957-GD). Recombinant GFP was purchased from Vector Laboratories (MB-0752). All oligonucleotides that were conjugated to antibodies, as well as the ligation splint, were purchased from Integrated DNA Technologies and were HPLC-purified. Oligonucleotide sequences are listed in Supplementary Table 5. Purified DNA-conjugated antibodies were quantified via the Bradford assay, using the Bio-Rad Protein Assay Dye Reagent Concentrate (5000006) with 2 mg/mL Pierce bovine gamma globulin standard ampules from Thermo Fisher Scientific (23212) as a standard. Dynabeads MyOne Streptavidin T1 (65601), 50 mM D-biotin (B20656), UltraPure salmon sperm DNA solution (15632011), UltraPure 0.5 M EDTA, pH 8.0 (15575020), nuclease-free water (not DEPC-treated) (AM9932), SYBR Green I nucleic acid gel stain (10,000X concentrate in DMSO) (S7563), and the Qubit dsDNA HS Assay Kit (Q32851) were purchased from Invitrogen. 10X phosphate-buffered saline (PBS) solution, Tween 20, and molecular-biology grade 200-proof ethanol were purchased from Fisher BioReagents (BP399 & BP337). Molecular biology-grade bovine serum albumin (BSA) was purchased from New England BioLabs (B9000S). Goat IgG was purchased from Millipore Sigma (I5256). Ampligase DNA Ligase and 10X Reaction Buffer were purchased from Lucigen (A3202K). 2X GoTaq G2 Hot Start Colorless Master Mix was purchased from Promega (9IM743). Uracil-DNA glycosylase (UDG) (1 U/μL) was purchased from Thermo Fisher Scientific (EN0362). Nextera XT Index Kits (96 indexes, 384 samples) were purchased from Illumina (FC-131-1002). Axygen AxyPrep Mag PCR Clean-up Kits were purchased from Thermo Fisher Scientific (MAGPCRCL). Buffer EB was purchased from Qiagen (19086). DynaMag-96 side magnets were purchased from Thermo Fisher Scientific (12331D). Viaflo 96-channel pipette and 300 μl pipette tips were purchased from Integra BioSciences (6432). All gel electrophoresis reagents were bought from Thermo Fisher Scientific: Novex 10% TBE (EC6875BOX), 6x loading dye (R0611), and O'RangeRuler 20 bp (SM1323). DNA quantification was done by Quant-iT ds DNA HS Assay (Q33120) from Thermo Fisher Scientific. Human serum samples were obtained from BioIVT (HUMANSRMUNN): male human serum, lot numbers HMN613433, HMN613434, and HMN613436, from donors aged 38, 38, and 45 years old, respectively. Luminex reagents were purchased from EMD Millipore.

### Detection antibody (dAb) functionalization

dAbs were generated by conjugating amine-modified oligonucleotides (15–25 $OD_{260}$ units) to polyclonal antibodies (100 μg) using the Antibody-Oligonucleotide All-in-One Conjugation Kit from TriLink Biotechnologies (A-9202), per the manufacturer's protocol. Oligos coupled to antibodies by their 3' end also featured 5' phosphorylation. Conjugations were performed according to the manufacturer's instructions, except that the coupling reactions were run overnight at 4 °C rather than for the recommended two hours at room temperature, as the oligonucleotides were longer (80–81 nucleotides) than the suggested maximum of 60 nucleotides recommended by the kit manufacturer. Conjugated probes were purified with TriLink Biotechnologies proprietary affinity magnetic beads provided in the kit as per the manufacturer's protocol.

### Solid-phase proximity ligation assay

The spPLA portion of the assay was run with only minor modifications from published protocols[17]. Capture antibody (cAb)-coated beads were prepared with MyOne Streptavidin T1 beads following the manufacturer's protocol. The stock beads (10 mg/ml) were resuspended on a rotator for 5 min. For each target, 50 μL beads were added to a 1.5-mL microcentrifuge tube. The tubes were placed on a magnet to pellet the beads, the storage buffer was removed, and the beads were washed three times with 200 μL wash buffer (1x PBS + 0.05% Tween 20). Biotinylated antibodies were reconstituted in storage buffer (1x PBS + 0.1% BSA) at 50 nM for all targets except CRP, which was reconstituted at 1.33 μM. After the third bead wash, 100 μL of biotinylated antibody solution was added to each tube. The beads were briefly vortexed to homogenize the solution, then incubated for 1 h at room temperature on a rotator. The tubes were then placed on a magnet, the supernatant was removed, and the beads were washed three times with 200 μL wash buffer. The beads were then resuspended in 100 μL storage buffer and used immediately or kept at 4 °C until used. CRP cAb beads were prepared using 2.5-fold greater volumes than indicated above, with the indicated higher antibody concentration for maximum binding capacity to reduce capture antibody saturation.

PLA buffer (1x PBS, 1 mg/mL BSA, 0.05% Tween 20, 15 μg/m goat IgG, 0.1 mg/mL salmon sperm DNA, and 5 mM EDTA) was prepared with GFP spiked in so that its final concentration in every sample would be 10 pM. GFP was used as an internal control to monitor variability in PLA steps, including ligation efficiency, bead washing, etc., as an exogenous analyte not found in human serum. Standard curves in GFP-spiked buffer were prepared by 3x serial dilution of a tube containing a mix of every target at its highest standard curve concentration. Non-target-spiked samples were prepared by simply using GFP-spiked buffer. For each sample, 40 μl of PLA buffer was added to a well in a 96-well plate.

Separately, the previously prepared cAb-coated beads (5 mg/mL) were homogenized via vortexing, and a 50 μL reaction was prepared in a tube containing 167 nL of 5 mg/mL bead solution ( ~700,000 beads) for each target. To avoid bead saturation, CRP required 10 times more beads than the other targets, or ~7,000,000 beads per 50 μL reaction. After adding each species of cAb-coated bead to a single tube, the tube was placed on a magnet, the storage buffer was removed, and the beads were resuspended in GFP-spiked PLA buffer to produce a solution in which each cAb-coupled bead species was present at 167 ng/mL (1.67 ug/mL for CRP). After mixing via gentle vortexing, 5 μL of this bead solution was added to each well in the PCR plate. Finally, 5 μl of buffer or undiluted chicken or human serum were added to each well. All samples were run in triplicate, with each sample having a final volume (after reagent addition) of 50 μL, and final GFP concentration of 10 pM. After sealing the plate, the samples were gently vortexed for 1 min and incubated on a rotator for 1.5 h at room temperature.

Following this incubation, the plate was spun down at 1000 rcf for 5 s and placed on a DynaMag 96-Well plate for 1 min to pellet the beads. The supernatant was removed, and 100 μL wash buffer added to each well using custom protocols on a Viaflo 96-channel pipette. The plate was again sealed, gently vortexed for 1 min, spun down at 1,000 rcf for 5 s, and placed on a magnet. The wash buffer was removed, and another wash performed as described above. Following the removal of the wash buffer from the second wash, each well was filled with 50 μL of PLA probe solution in non-GFP-spiked PLA buffer, containing each target's probes at their optimized concentrations (Supplementary

Table 2), as well as any unlabeled polyclonal antibodies used as an epitope depletant. The plate was sealed, vortexed gently for 1 min, and incubated for 1.5 h on a rotator at room temperature.

The plate was again spun down and placed on the magnet. The supernatant from each well was removed with the Viaflo, and the wells were washed two times with 100 µL wash buffer as described in the previous paragraph. Following the second wash, each well was filled with 50 µL of the ligation reaction solution, consisting of 0.05 U/µL Ampligase, 100 nM ligation splint, 1X Ampligase reaction buffer, and nuclease-free water. The plate was again sealed, gently vortexed for 1 min, spun down at 1000 rcf for 30 s, and placed in a thermocycler and incubated at 50 °C for 10 min to allow the ligation reaction to occur. Following this reaction, the plate was spun down at 1000 rcf for 1 min and placed on a magnet. The supernatant was removed, and the wells were again washed twice with 100 µL wash buffer.

Following the second wash, 40 µL of PCR reaction solution was added to each well, consisting of 1x GoTaq G2 Hot Start Colorless Master Mix, 0.02 U/µl UDG, 100 aM control oligo, and PCR-grade water. Samples were placed at 4 °C overnight to continue the protocol the next day. To each sample, we added 5 µL of each of two different Nextera XT indices (10X stock). These indexes act both as PCR primers to amplify the DNA as well as sample indices to uniquely identify each sample, as the pair of indices added to each well in a PLA-Seq run will be unique to that run. This enables the pooling of all the samples from a single PLA-Seq run (up to 384 samples with the original index kit, or higher if combining index kits). Following sequencing, these are de-multiplexed based on their unique index pair.

Following the addition of the indices, the plate was sealed, gently vortexed, spun down for 1 min at 1000 rcf, and placed in a thermo-cycler for a preamplification reaction: 10 min at 95 °C (to activate the polymerase and deactivate the UDG), 72 °C for 3 min, 95 °C for 30 s, and 4 cycles of 95 °C for 10 s, 55 °C for 30 s, and 72 °C for 90 s, followed by 5 min at 72 °C and 5 min at 4 °C. The plate was then spun down at 1,000 rcf for 1 min and placed on a magnet. From each 50 µL reaction, 33 µL was removed and transferred to a new PCR tube, which was stored at 4 °C. To the remaining 17 µL/well, we added 2 µL 10X SYBR Green I dye. The plate was covered, gently vortexed for 1 min, and spun down for 1 min at 1,000 rcf. The plate was then analyzed via qPCR to determine the number of cycles that each sample (*i.e.*, the 33 µL in the PCR tubes) should be amplified so that all samples produce roughly the same amount of total DNA. The qPCR protocol was: 72 °C for 3 min, 95 °C for 30 s, and 39 cycles of 95 °C for 10 s, 55 °C for 30 s, and 72 °C for 30 s. To ensure that every sample is amplified to roughly the same DNA output, we calculated the number of cycles required for each sample to reach 0.25 maximum fluorescent value (Ct).

Ct values were extracted with a custom python script. Each amplification intensity value ($I_i$) across all 39 cycles was normalized by:

$$I_{i\_norm} = \frac{I_i - \min(I)}{\max(I) - \min(I)} \quad (1)$$

Using these normalized values, the cycle that first passes the threshold value of 0.25 was the Ct value.

This value was then used as the number of amplification cycles for the remaining 33 µL of sample. If samples varied by <3 cycles, all samples were amplified to 1 + average Ct. The amplification protocol itself was identical to the qPCR protocol (without plate reads); after amplification was complete, samples were removed during the 90 s 72 °C step and incubated in another thermocycler at 72 °C for an additional 5 min before being removed and placed at room tempera-ture. Once every sample was amplified, we performed PCR cleanup using an Axygen AxyPrep Mag PCR Clean-up Kit based on the manu-facturer's instructions, using the Viaflo multi-channel pipette for high throughput. The resulting purified PCR products were then quantified using the Quant-iT dsDNA protocol. Samples were then pooled together at equal molar concentrations. The samples were quality-controlled by native gel electrophoresis at 180 V for 40 min. The pool was then quantified again with Qubit and sent for sequencing on an Illumina MiSeq at the Stanford Functional Genomics Facility.

For a qPCR readout (as in the tuning experiments), normal PLA buffer was used throughout the above protocol instead of GFP-spiked PLA buffer. Additionally, after the wash following ligation, 50 µl of the following PCR mix was added to the beads: 1x GoTaq G2 Hot Start Colorless Master Mix, 0.02 U/µl UDG, 0.25x SYBR Green, 500 nM Uni-versal Forward Primer, 500 nM Universal Reverse Primer, and PCR-grade water. For the qPCR readout, the protocol above ended with the Cq extraction.

## DNA calibration curves for qPCR
To convert qPCR Ct values to approximate DNA concentrations, we first made a calibration curve with a serial dilution of the full sequence of the IL-1ra reporter. We conducted qPCR measurements in triplicate for 100, 10, 1, 0.1, 0.01, and 0 fM DNA, and calculated Ct values. Since the Ct values were highly linear with log(input DNA template), we fitted the following equation:

$$Ct = m * \log([DNA]) + b \quad (2)$$

where Ct is the calculated 0.25 max value obtained from qPCR amplification of the input template DNA concentration. The para-meters m and b were determined for all s conversions from Cq to [DNA] as follows:

$$[DNA] = 10^{\frac{Cq-b}{m}} \quad (3)$$

## Data extraction analysis
FASTQ files were generated using FASTQ Generation v1.1.0. FASTQ files were analyzed using custom code written in Perl. Code is available at https://github.com/newmanst/evros. After demultiplexing in Base-Space according to Illumina indices, reads were filtered by quality; only reads 52 nt in length with Phred scores ≥20 for all bases were used in our analysis. Duplicate reads were compiled and counted using the *fastaptamer_count* v1 package[25]. After each UMI was only represented once per file, the compiled reads were aligned to the known reporter sequences (Supplementary Table 5). An exact similarity score was used, as opposed to Needleman Wunsch or SmithWaterman, because the split regions contain high degrees of similarity shifted by a few bases. An exact similarity score minimizes false positives by empha-sizing the sequence similarity in the protein tag region. Each read was assigned to the reporter with the highest similarity score, provided the similarity score was greater than 31. Reads that returned similarity scores <31 for all reporters were discarded. The resulting output comprises the total number of unique barcodes for every target reporter.

Each sample also included an internal control reporter oligo to reduce intra-assay variance that arises from variations in experimental factors such as ligation efficiency, library prep, and pooling for sequencing. Outputs were reported in terms of normalized unique molecular identifiers (nUMI), which reflects the number of UMIs for the target divided by the number of UMIs for the control reporter oligo.

## Curve fitting and quantification
Curve fitting and quantification were done with a custom python script. To create a calibration curve, we fitted our data to the Four-Parameter Logistic Curve (4-PL),

$$nUMI = \frac{A - D}{1 + \left(\frac{x}{C}\right)^B} + D \quad (4)$$

where nUMI is the normalized counts from sequencing, x is the analyte concentration, A is the minimum value possible with no analyte, B is the Hill coefficient, C is the point of inflection ($K_d$), and D is the maximum value possible with the infinite analyte. The parameters A, B, C, and D were determined by the curve-fitting function in python using Scipy's optimize curve fit function, which uses non-linear least squares to fit a function. We used the log residual for the loss function. To calculate the concentration for a given signal, we then used the inverse function with fit parameters A, B, C, and D as follows:

$$x = C\left(\frac{A-D}{nUMI-D}-1\right)^{1/B} \tag{5}$$

## Luminex sample prep

Three separate Luminex panels were used for the four targets according to the manufacturer's protocols: Human Cardiovascular 2 MAG (GDF-15; Cat# MXHCV2M0N02189), Neurodegenerative MAG Panel 2 (CRP; Cat# MXHNDG2M0N02068), and Human Cyto/Chem/GF Panel A (IL-6, IL1-RA; Cat# MXHCYTA0N03031). The recommended sample dilutions for each panel (3x, 100x, and 1,000x, respectively) were prepared with 1x PBS. Kit-provided standards were prepared with four-fold serial dilutions with assay buffer. For each sample, 25 μl of buffer or matrix (per manufacturer's recommendation—buffer for CRP and GDF-15 panels, and matrix for IL-1ra/IL-6 panels) was added to 25 μl of magnetic beads and 25 μl of the sample at the recommended dilution or standard. Samples were incubated for 16 h at 4 °C on a shaker. After incubation, the samples were washed in a magnetic washer and 25 μl of detection reagent was added and incubated for 1 h. SAPE was added for 30 min at room temperature on a shaker, and the samples were then washed on a magnetic washer. After adding 130 μl of reading or wash buffer, the samples were placed on a shaker for 3 min, then placed in a Luminex 100/200 System for reading. All samples were tested in duplicate and analyzed with standard quantification procedures recommended by Luminex.

The dilutions spanned a three-log range: no dilution, 3-fold, 10-fold, 100-fold, 1,000-fold, and 2,000-fold. Serum samples were diluted in PBS in a master plate and applied to all four panels. All sample dilutions were analyzed on all three panels. 25 μl of the diluted sample was mixed with antibody-linked magnetic beads in a 96-well plate and incubated overnight at 4 °C with shaking. Cold and room temperature incubation steps were performed on an orbital shaker at 500–600 rpm. Plates were washed twice with wash buffer in a BioTek ELx405 washer. Following 1 h incubation at room temperature with biotinylated detection antibody, streptavidin-PE was added for 30 min with shaking. Plates were washed as described above and PBS was added to wells for reading in the Luminex FlexMap3D Instrument with a lower bound of 50 beads per sample per cytokine. Each sample was measured with two replicates. Wells with a bead count <50 were flagged, and wells with a bead count < 20 were excluded.

## Statistics & reproducibility

All samples except Luminex data were done in triplicate and the mean and standard deviations were calculated. Luminex measurements were done in duplicates due to limited throughput and large sample volume required. All attempts at replication are presented in the figure plots. No statistical method was used to predetermine the sample size. The experiments were not randomized. Sequence processing was blinded as samples were sent out to the Functional Genomics Facility for sequencing. For all other data, the Investigators were not blinded to allocation during experiments and outcome assessment.

## Reporting summary

Further information on research design is available in the Nature Portfolio Reporting Summary linked to this article.

## Data availability

The data generated in this study have been deposited and are publicly available in the Stanford Digital Repository (https://purl.stanford.edu/pf688zn8684). Source data are also provided with this paper. Source data are provided with this paper.

## Code availability

Code is available on github (https://github.com/newmanst/evros)[26]

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

## Acknowledgements
S.S.N., B.D.W., D.M., H.N. were supported by Genentech Grant # CR-0000579. S.S.N acknowledges support from SGF (Stanford Graduate Fellowship in Science & Engineering) and the NSF Graduate Research Fellowship Program (GRFP)

## Author contributions
H.T.S., H.T.M., B.D.W., D.M., and S.S.N. conceived the project. S.S.N., B.D.W, and D.M., designed the experiments. S.S.N. executed the experiments with the help of D.M., B.D.W., B.C.W. H.N. Y.R. executed the Luminex experiments. S.S.N and B.D.W. developed the code and analyzed the data with the help of B.C.W. S.S.N., B.D.W., D.M., B.C.W., M.E., and H.T.S wrote the manuscript. All authors reviewed and approved the final manuscript.

## Competing interests
H.T.S., B.D.W., S.S.N., and D.M. are inventors on a filed patent application to the US patent office pertaining to the molecular-equalization aspect of this work (application number # 63/309,981). The remaining authors declare no competing interests.
