## [Peer Review File · Nature Communications]

REVIEWER COMMENTS

Reviewer #1 (Remarks to the Author):

This well written and executed work. As far as I understand the conclusions are sound. The problem with non-linearity of dilution of serum and plasma samples is well known and is seen by immunoassay experts as a sign of a not fully optimized assay. Very well optimized assays will not exhibit non-linearity when diluted. But this is difficult to get consistent especially in multiplex where all assays operate under the same buffer conditions.

The only concern about the paper is that the described approach of epitope depletion and probe loading is already common practice in the immunoassay industry. However, it has not been published before (as far as I know) so that should not prohibit publication and the concepts would be of interest to a wider audience. But calling it a new method with a cool name might be overselling it.

In figure 2e, the probe loading process usually increases the background signal of the assay which is not shown in the theoretical figure. This effect is fine, as the detection is for a very abundant protein.

I recommend publication. The work can be reproduced from the descriptions.

Reviewer #2 (Remarks to the Author):

The paper "Extending the dynamic range of biomarker quantification through molecular equalization" by Newman et al. reports an interesting approach to simultaneously measure proteins present at widely different expression levels in a sample, using solid-phase proximity ligation assays, by adjusting the output for each protein. Two simple factors are adjusted to achieve the desired reporting rate: the concentrations of oligonucleotide-conjugated target-specific antibodies, and of unconjugated versions of such antibodies. These are two parameters that are well known to influence output of affinity-based protein assays, but this paper gives a clear, informative and helpful mechanistic description how to

arrive at the desired correction of detection reactions, and the presented data convincingly support the conclusions. The information in this paper can be helpful for setting up several forms of multiplex protein assays. Some specific points are as follows:

1. The authors appropriately refer to their correction mechanism as the EVROS strategy in the Abstract on page 2 and as an equalization methodology in Conclusion on page 15. However, somewhat confusingly they mention “our EVROS assay” at the top of page 5 and “the EVROS assay” at the bottom of page 10, and again in Figure 5 on page 15 they seem to refer to EVROS as an assay mechanism by comparing it to Luminex. It would be better to stick to the definition of EVROS as a correction mechanism throughout, rather than to describe it as an assay mechanism.
2. At the end of the Introduction on page 4 the relevance of the EVROS approach for various proximity assays is mentioned, and again in the Conclusion at the bottom of page 15 the EVROS technique is described as also being applicable to any other proximity-based assay. However, at the bottom of page 5 the authors mention that the two tuning mechanisms could be applied to many types of immunoassays besides proximity assays as seems plausible. Maybe this point about possible applications for the EVROS mechanism in other assays can be clarified in the paper.
3. Table 1 could preferably be moved to Supplemental Information.
4. Figure 4a on page 13 gives the output also in the absence of the four target proteins, which is helpful. Was this background subtracted from the signals in Figure 4b? This is important in order to evaluate the y-intercept reported at the bottom of page 12.
5. If output values are available for zero concentrations of the various targets shown in Figure 3 this should preferably be included in that figure too.
6. The authors suggest that this may be the first time this wide range of endogenous concentrations are handled in an assay thanks to their correction mechanism. This may be true but it could be useful to look at the high-multiplex panels of assays offered by Somalogic and Olink Proteomics to see how their investigated concentration ranges compare with those presented in this paper.
7. The legend of Figure 5 on page 15 describes error bars but these are not visible in the figure. Instead all replicate values have been plotted, which I find preferable. The horizontal line presumably indicating the average value of a triplicate for the IL-6 reaction called EVROS H2 seems off.
8. In the Materials and Methods section on page 21 it could be useful to mention if the antibody-oligonucleotide conjugates were purified, even if that information is available in the manufacturers protocol, since any remaining unconjugated antibodies or oligonucleotides would be expected to interfere with the EVROS correction mechanisms.
9. The sliding splint shown under supplemental Information, page 28, as well as the HTS readout of spPLA reactions performed with polyclonal antibody preparations split three ways as used in this paper seems to be inspired by a paper by Darmanis et al. PLoS One 6: e25583, 2011 and this could therefore be a useful reference to include.

Reviewer #3 (Remarks to the Author):

The manuscript by Newman et al. described a strategy to extend the dynamic range for quantifying biomarkers in biofluidic samples. This study covers a highly important yet often overlooked issue in analyzing biomarkers in clinical and biological samples, the dynamic range. The authors correctly identified the issue of the limited dynamic range of existing analytical assays. They proposed a molecular equalization strategy for extending the dynamic range for quantifying multiple biomarkers within a wide 10^7 concentration range. The proposed strategy includes a probe loading for enhancing the detection of analytes in a low concentration range and an epitope depletion approach for analytes at the high concentration range. This molecular equalization strategy is interesting and practically highly useful. My comments lie mainly in some of the claims and technical details:

- 1) The authors raised the issue of the nonlinear dilution effect. However, it is not clear how this nonlinear dilution effect would influence the PLA assay used in the present study. I would recommend the authors perform a head-to-head comparison between EVROS and sample dilution so that one can clearly understand the nonlinear dilution effect and appreciate the necessity of EVROS.
- 2) While the epitope depletion approach was straightforward for helping suppress the saturated detection signals by high concentrations of analytes, the probe loading approach was not so straightforward for two reasons: first, it seems that the probe loading had very little effect at the low concentration range in Figure 2 c, so not sure how this would help enhance the detection of low concentration analyte. More discussion shall be given to clarify that.
- 3) The second reason was when raising the concentration of probes, a higher background signal was often expected. This is probably why all PLA assays need to optimize the probe concentrations to achieve maximal analytical performance. How was the background issue addressed when using probe loading in EVROS?
- 4) To use EVROS, it is critical to gain prior knowledge of the concentration range of biomarkers. For analytes without prior knowledge, can EVROS still work? And how?
- 5) Can EVROS be generalized to other types of immunoassays beyond proximity assays, such as ELISA or CLIA?
- 6) A cartoon to clearly demonstrate the idea and workflow of EVROS shall be created and placed at the very beginning of the results section, which would give a better idea of how EVROS works.
- 7) The comparison between EVROS and Luminex assay was not quite clear, a better discussion shall be made on how each assay, especially EVROS was performed when comparing against Luminex.

Reviewer 1:

The reviewer praised our work as “well written and executed”, and we greatly appreciate their positive reception of our manuscript. However, they also raised a few points for clarification, which we have addressed below.

- 1. The only concern about the paper is that the described approach of epitope depletion and probe loading is already common practice in the immunoassay industry. However, it has not been published before (as far as I know) so that should not prohibit publication and the concepts would be of interest to a wider audience. But calling it a new method with a cool name might be overselling it.**

We thank the reviewer for pointing this out—indeed, as academics we were not aware of these approaches being existing practices, as we had not seen such strategies described in the literature. We have added a sentence to the Conclusion to address this point. However, we also feel that as the first published description of the systematic implementation of this approach in the research literature—and likely the first encounter of this approach for many academic labs—it is appropriate to formulate it as a new method and suggest a descriptive name for this strategy.

- 2. In figure 2e, the probe loading process usually increases the background signal of the assay which is not shown in the theoretical figure. This effect is fine, as the detection is for a very abundant protein.**

The reviewer is correct, and we did not depict this effect in **Figure 2e** to minimize confusion from the main point of probe loading. However, to show that this effect does happen, we have now explicitly plotted actual background signal versus probe concentration as **SI Figure 5**; these data were originally shown in **Figure 4a** at 0 concentration.

Reviewer #2:

We thank the reviewer for their thoughtful comments, suggestions, and support of our work – it is greatly appreciated. We have addressed all the concerns as shown below.

- 1. The authors appropriately refer to their correction mechanism as the EVROS strategy in the Abstract on page 2 and as an equalization methodology in Conclusion on page 15. However, somewhat confusingly they mention “our EVROS assay” at the top of page 5 and “the EVROS assay” at the bottom of page 10, and again in Figure 5 on page 15 they seem to refer to EVROS as an assay mechanism by comparing it to Luminex. It would be better to stick to the definition of EVROS as a correction mechanism throughout, rather than to describe it as an assay mechanism.**

We apologize for the confusion and have updated the manuscript so that EVROS only refers to the equalization methodology.

- 2. At the end of the Introduction on page 4 the relevance of the EVROS approach for various proximity assays is mentioned, and again in the Conclusion at the bottom of page 15 the EVROS technique is described as also being applicable to any other proximity-based assay. However, at the bottom of page 5 the authors mention that the two tuning mechanisms could be applied to many types of immunoassays besides proximity assays as seems plausible. Maybe this point about possible applications for the EVROS mechanism in other assays can be clarified in the paper.**

Although this approach may be compatible with a broader range of immunoassays, our intended meaning was that the principles outlined in the paper can specifically be applied to other types of proximity-based immunoassays. We have updated the wording accordingly.

- 3. Table 1 could preferably be moved to Supplemental Information.**

We agree with the reviewer and have updated the manuscript accordingly.

- 4. Figure 4a on page 13 gives the output also in the absence of the four target proteins, which is helpful. Was this background subtracted from the signals in Figure 4b? This is important in order to evaluate the y-intercept reported at the bottom of page 12.**

The signal data plotted in **Figure 4a** and **4b** are the same, and no background subtraction was applied in either case. The background is not plotted in **Figure 4b**, as these signals were not used for the fit referred to at the bottom of page 12. We have updated the figure caption to reflect this.

- 5. If output values are available for zero concentrations of the various targets shown in Figure 3 this should preferably be included in that figure too.**

The focus of **Figure 3** was not on the probe concentration-related increase in background signal, and we only ran the highest probe concentration as a control in the qPCR tuning experiments to ensure that the background signal was not higher than the overall signal output. To highlight the background signals for different probe concentrations, we have generated a plot of the background

signals produced by the final probe concentrations used for **Figure 4a** (multiplexed quantification). This figure is provided in response #2 to Reviewer #1 above and is also included in the revised manuscript as **SI Figure 5**.

- 6. The authors suggest that this may be the first time this wide range of endogenous concentrations are handled in an assay thanks to their correction mechanism. This may be true but it could be useful to look at the high-multiplex panels of assays offered by Somalogic and Olink Proteomics to see how their investigated concentration ranges compare with those presented in this paper.**

We have updated our language to clarify that we are, for the first time, measuring this wide range of concentrations without differential dilution in a single measurement. Commercial platforms such as Somalogic and Olink can achieve a similar dynamic range by using differential dilution and across multiple panels/measurements.

- 7. The legend of Figure 5 on page 15 describes error bars but these are not visible in the figure. Instead all replicate values have been plotted, which I find preferable. The horizontal line presumably indicating the average value of a triplicate for the IL-6 reaction called EVROS H2 seems off.**

We have updated the caption for **Figure 5** to describe the data presentation more accurately. The position of the horizontal line originally marked the median value, and was thus correct for the triplicate measurement for IL-6 in EVROS H2. For clarity, we have updated the positioning of the horizontal lines in this figure to indicate average values.

- 8. In the Materials and Methods section on page 21 it could be useful to mention if the antibody-oligonucleotide conjugates were purified, even if that information is available in the manufacturers protocol, since any remaining unconjugated antibodies or oligonucleotides would be expected to interfere with the EVROS correction mechanisms.**

We thank the reviewer for reminding us to highlight this important detail. This was described in the manufacturer's protocol, and we have updated the Methods section to summarize the purification method, which made use of TriLink Biotechnologies' proprietary affinity magnetic beads.

- 9. The sliding splint shown under supplemental Information, page 28, as well as the HTS readout of spPLA reactions performed with polyclonal antibody preparations split three ways as used in this paper seems to be inspired by a paper by Darmanis et al. PLoS One 6: e25583, 2011 and this could therefore be a useful reference to include.**

We were indeed inspired by multiple works from this research group—we cited Darmanis *et al.* (*Mol Cell Proteomics* **9**, 327, 2010) as a reference for solid-phase PLA in our Introduction, as well as the work from Nong *et al.* (*Nat. Protoc.*, **8**, 1234, 2013) for spPLA, which describes the splitting of the polyclonal antibodies, the use of a sliding splint for multiplexing, and the sequencing readout. For extra clarity, we have also added the suggested 2011 paper as Ref. #7.

Reviewer #3:

The reviewer was generally positive about our work and commented “this molecular equalization strategy is interesting and practically highly useful.” We greatly appreciate their support. We have addressed the technical questions raised by the reviewer as shown below.

- 1) **The authors raised the issue of the nonlinear dilution effect. However, it is not clear how this nonlinear dilution effect would influence the PLA assay used in the present study. I would recommend the authors perform a head-to-head comparison between EVROS and sample dilution so that one can clearly understand the nonlinear dilution effect and appreciate the necessity of EVROS.**

To address the reviewer’s question, we offer data from an experiment in which a known amount of analyte was spiked into three human serum samples and diluted 3-, 9-, or 27-fold with PLA buffer, and then measured for analyte recovery efficiency using both the spPLA and HTS readouts. The results are shown below:

Each color represents a different sample. The green shaded region depicts a generous +/- 20% error range for acceptable levels of linearity in dilution. As expected, the recovery was non-linear across dilutions and the magnitude of this nonlinearity varied both for different analytes and different samples. CRP was highly divergent in terms of adjusted recovery at just 27-fold dilution, whereas conventional assays for this analyte typically entail >1,000-fold dilution. These data are presented in the revised manuscript as **SI Figure 8**.

- 2) **While the epitope depletion approach was straightforward for helping suppress the saturated detection signals by high concentrations of analytes, the probe loading approach was not so straightforward for two reasons: first, it seems that the probe**

loading had very little effect at the low concentration range in Figure 2c, so not sure how this would help enhance the detection of low concentration analyte. More discussion shall be given to clarify that.

We thank the reviewer for pointing this out. Below, we have magnified to the lower concentrations plotted in **Figure 2c**:

At this scale, one can clearly see the increase in signal output that comes with increasing probe concentrations. We would like to stress that this model was solely intended to predict the behavior resulting from parameter tuning and to show the theoretical possibility of decoupling signal output from antibody affinity and not the exact scale of the effect. We confirmed experimentally that probe loading indeed increases the signal output in **Figure 3a**.

- 3) **The second reason was when raising the concentration of probes, a higher background signal was often expected. This is probably why all PLA assays need to optimize the probe concentrations to achieve maximal analytical performance. How was the background issue addressed when using probe loading in EVROS?**

The reviewer is correct that the background signal from spPLA increases with probe loading, as we have shown in **SI Figure 5** (see above, in response #2 to Reviewer #1). Importantly, we maintain a high SNR despite having higher background with the lower-concentration analytes for which higher levels of probe loading are required. We would also like to emphasize that the purpose of this work is not to maximize assay sensitivity per se, but rather to achieve multiplexed measurement of clinically-relevant analyte concentrations across a large dynamic range, which we believe we have successfully demonstrated. We would also like to note that the spPLA protocol includes many wash steps that inherently decrease the background signal resulting from the use of additional probe. Other assays may require further optimization if the SNR is not sufficiently high.

- 4) **To use EVROS, it is critical to gain prior knowledge of the concentration range of biomarkers. For analytes without prior knowledge, can EVROS still work? And how?**

The reviewer is correct that as presented in this work, EVROS requires prior knowledge of the expected concentration range of the biomarker.

- 5) **Can EVROS be generalized to other types of immunoassays beyond proximity assays, such as ELISA or CLIA?**

It is possible that this tuning strategy could be generalized to other immunoassay categories, but the present work is focused entirely on proximity-based assays. We have updated the wording of the manuscript to make this clearer.

- 6) A cartoon to clearly demonstrate the idea and workflow of EVROS shall be created and placed at the very beginning of the results section, which would give a better idea of how EVROS works.

For clarity, we have illustrated the workflow for proteinSeq and spPLA—the assay we have employed in this demonstration of EVROS—in SI Figure S2 (shown below). We include this figure in the SI rather than the main text to keep the reader focused on EVROS as a strategy that is not wedded to this specific assay format. As noted above, the EVROS strategy can be implemented with any proximity-based immunoassay.

- 7) The comparison between EVROS and Luminex assay was not quite clear, a better discussion shall be made on how each assay, especially EVROS was performed when comparing against Luminex.

We have updated the wording of this section and elaborated how the EVROS tuning mechanism was applied to spPLA and how these results were compared to those obtained from Luminex.

REVIEWERS' COMMENTS

Reviewer #1 (Remarks to the Author):

The authors have addressed all my suggestions well. Looking forward to the publication.

Reviewer #2 (Remarks to the Author):

The authors have satisfactorily addressed the points I raised and in my view the paper can now be published.

Reviewer #3 (Remarks to the Author):

With revisions made by the authors, the manuscript has been significantly improved. I believe it is acceptable in its present form.

Review Round 2:

We thank all reviewers for their favorable recommendations and acceptance of our response and revisions in review round 1.

Reviewer #1:

The authors have addressed all my suggestions well. Looking forward to the publication.

Reviewer #2:

The authors have satisfactorily addressed the points I raised and in my view the paper can now be published.

Reviewer #3:

With revisions made by the authors, the manuscript has been significantly improved. I believe it is acceptable in its present form.